# The Protective Effect of Carotenoids, Polyphenols, and Estradiol on Dermal Fibroblasts under Oxidative Stress

**DOI:** 10.3390/antiox10122023

**Published:** 2021-12-20

**Authors:** Aya Darawsha, Aviram Trachtenberg, Joseph Levy, Yoav Sharoni

**Affiliations:** Department of Clinical Biochemistry and Pharmacology, Faculty of Health Sciences, Ben-Gurion University of the Negev, Beer Sheva 8410501, Israel; ayadar@post.bgu.ac.il (A.D.); aviramtr@post.bgu.ac.il (A.T.)

**Keywords:** lycopene, carnosic acid, apoptosis, matrix metalloproteinase (MMP), collagen, antioxidant response element/Nrf2 (ARE/Nrf2), reactive oxygen species (ROS), NHDF cells

## Abstract

Skin ageing is influenced by several factors including environmental exposure and hormonal changes. Reactive oxygen species (ROS), which mediate many of the effects of these factors, induce inflammatory processes in the skin and increase the production of matrix metalloproteinases (MMPs) in dermal fibroblasts, which leads to collagen degradation. Several studies have shown the protective role of estrogens and a diet rich in fruits and vegetables on skin physiology. Previous studies have shown that dietary carotenoids and polyphenols activate the cell’s antioxidant defense system by increasing antioxidant response element/Nrf2 (ARE/Nrf2) transcriptional activity and reducing the inflammatory response. The aim of the current study was to examine the protective effect of such dietary-derived compounds and estradiol on dermal fibroblasts under oxidative stress induced by H_2_O_2_. Human dermal fibroblasts were used to study the effect of H_2_O_2_ on cell number and apoptosis, MMP-1, and pro-collagen secretion as markers of skin damage. Treatment of cells with H_2_O_2_ led to cell death, increased secretion of MMP-1, and decreased pro-collagen secretion. Pre-treatment with tomato and rosemary extracts, and with estradiol, reversed the effects of the oxidative stress. This was associated with a reduction in intracellular ROS levels, probably through the measured increased activity of ARE/Nrf2. Conclusions: This study indicates that carotenoids, polyphenols, and estradiol protect dermal fibroblasts from oxidative stress-induced damage through a reduction in ROS levels.

## 1. Introduction

The skin is affected by environmental factors such as exposure to UV radiation, air pollutants, xenobiotics, and cigarette smoke [1,2], and by endogenous changes that occur with ageing [3]. These factors generally increase oxidative stress by generating reactive oxygen species (ROS) [1,2]. Oxidative stress, defined as a disturbance in the balance between the production of ROS and the antioxidant defenses [4], leads to changes in DNA, proteins, and lipids; and activation of inflammatory processes and signaling pathways that may cause various chronic and degenerative diseases [5], such as diabetes [4], cancer [6], neurological [7], and cardiovascular diseases [8]. Oxidative stress can also cause skin damage due to induction of apoptotic cell death [9,10]; activation of inflammatory processes [1]; an increase in metalloproteinases (MMPs), which degrade collagen in the extracellular matrix [11,12]; and a reduction in collagen production [13]. All of these changes accelerate skin ageing. 

Epidemiological and clinical studies indicate that a diet rich in fruits and vegetables improves skin health [14], and that plant-derived compounds (phytonutrients) reduce oxidative stress-induced skin damage [15,16]. Among these active phytonutrients are carotenoids and polyphenols [16]. It was suggested that the protective effects of phytonutrients are mediated by the modulation of signaling pathways that are involved in the adverse effects of oxidative stress. This modulation includes reduction in the activities of the nuclear factor-κB (NF-κB) transcription system, activator protein 1 (AP-1), and mitogen-activated protein kinase (MAPK) [17,18], and improving the antioxidant defense mechanism [19]. Several studies, including our own, suggest that modulation of these signaling pathways by phytonutrients can lead to a reduction in inflammatory processes [20,21,22] and MMP expression [17,23], and increase collagen synthesis [17]. 

Estradiol plays an important role in maintaining skin health [18], which is impaired in menopause due to a dramatic reduction in estradiol levels [18]. Treatment with estrogens (hormonal replacement therapy) increases collagen levels which improve skin elasticity [24]. These effects are partially associated with a reduction in oxidative stress [18] through increasing the expression of antioxidant enzymes and modulating the activity of the antioxidant response element/Nrf2 (ARE/Nrf2) transcription system [25]. Carotenoids were shown to activate the ARE/Nrf2 transcription system in several cell types [26], including keratinocytes [21], the main cell type of the skin epidermal layer. This system is also activated by polyphenols [21,27]. The activation of ARE/Nrf2 by carotenoids is probably exerted by carotenoid-oxidized derivatives which can be formed in the cells, but not by the intact molecules [28].

The aim of the current study was to investigate the effect of oxidative stress on primary human dermal fibroblasts and the ability of phytonutrients and estradiol to protect the cells from ROS-induced damage. These fibroblasts, which originate in the dermis layer of the skin, were used because they are the main source of collagen in the skin. Incubation with H_2_O_2_ was used to increase intracellular ROS. The effect of both H_2_O_2_ and the tested compounds (tomato and rosemary extracts and estradiol) was examined on cell death, MMP-1 and pro-collagen 1a1 secretions, and ROS level. The combined effect of H_2_O_2_ and the tested compounds on the activation of the ARE/Nrf2 transcription system was examined as a possible mechanism for the protection of dermal fibroblasts from oxidative stress.

## 2. Materials and Methods

### 2.1. Materials

Tomato extract (Lycomato^TM^) and rosemary extract were a gift of Lycored Ltd., Beer Sheva, Israel). The tomato extract, prepared by ethyl acetate extraction of tomato pulp, contained 6% lycopene, other tomato carotenoids (phytoene and phytofluene above 1%, beta-carotene above 0.2%), and additional fat-soluble tomato components such as natural tocopherols (above 1.5%) and phytosterols (1.1–2.5%). The remainder were triacylglycerols (70–72%), monoacylglycerols (8–9%), and phospholipids (7–8%). The rosemary extract was prepared by extraction with 80% ethanol. Its composition was only partially determined to contain carnosic acid (20.2%) and carnosol (2.5%) as the main polyphenols. 17β-estradiol was purchased from Sigma-Aldrich (Rehovot, Israel). Carotenoids were dissolved in tetrahydrofuran (THF), solubilized in cell culture medium, and their final concentration was measured as described previously [28,29,30]. Rosemary extract and estradiol were dissolved in ethanol. H_2_O_2_ 30% and THF, containing 0.025% butylated hydroxytoluene as an antioxidant, were purchased from Sigma-Aldrich. Dulbecco’s modified Eagle’s medium (DMEM), dextran-coated charcoal-treated fetal bovine serum (DCC-FBS), Hanks’ solution, and 1M HEPES buffer were purchased from Biological Industries (Beth Haemek, Israel). 

### 2.2. Cell Culture

Normal human dermal fibroblasts (NHDF) were purchased from PromoCell GmbH (Heidelberg, Germany). The cells were grown in PromoCell fibroblast growth medium 2, according to the manufacturer’s instructions, in a humidified atmosphere at 37 °C in 5% CO_2_. Before each experiment, cells were depleted of steroid hormones by maintaining them for 3–5 days in phenol red-free DMEM supplemented with 10% DCC-FBS (DMEM-DCC-FBS). This medium was used throughout all experiments because it does not contain steroid hormones or any other compound with estrogenic activity, such as phenol red.

### 2.3. Determination of Cell Number, and Secretion of MMP-1 and Procollagen 1a1

NHDF cells were seeded in 96-well plates at a density of 10^4^ cells/well in DMEM-DCC-FBS medium. Twenty-four hours later, cells were pre-incubated with phytonutrients or estradiol for 24 h. Vehicle-treated control cells were incubated with the relevant amounts of solvents used in a particular experiment, which had no effect on the measured parameters. The medium was then replaced with one containing the treatment compounds, with or without H_2_O_2_ and incubated for an additional 24 h. Thereafter, medium was removed and frozen for the analysis of secreted proteins, and the cell number was determined by the XTT Cell Proliferation Kit (Biological Industries, Beth Haemek, Israel), according to the manufacturer’s instructions. MMP-1 and pro-collagen 1a1 protein levels in cell culture supernatants were quantified by ELISA using the Human Total MMP-1 DuoSet and Human Pro-Collagen 1a1 DuoSet, ELISA kits (R&D Systems, Inc., Minneapolis, MN, USA), according to the manufacturer’s instructions. Optical density was measured using a VERSAmax tunable microplate reader (Molecular Devices, Menlo Park, CA, USA). Results of the cell number, MMP-1, and pro-collagen were calculated as a percent of the values obtained in control cells, treated with a vehicle without H_2_O_2_.

### 2.4. Assessment of Apoptosis 

Cells were seeded in 6-well plates at 3 × 10^5^ cells/well. Twenty-four hours later, they were pre-incubated with the phytonutrients or estradiol for 24 h. The medium was then replaced with one containing the treatment compounds, with or without H_2_O_2_, and incubated for an additional 12 h. Cells were washed in PBS and stained with Annexin V and 7-aminoactinomycin D (7-AAD), using the Annexin V-FITC/7-AAD Apoptosis Detection Kit (Biogems, Cat# 62700-50, Chai Wan, Hong Kong), according to the manufacturer’s protocol. The percentages of apoptotic cells were determined by flow cytometry on a Gallios instrument (Beckman Coulter Gallios Flow Cytometer, Brea, CA, USA). For each analysis, 10,000 events were recorded, and the data were processed using Kaluza software, version #2.1, (Gallios^TM^ Kaluza). As a positive control, cells were incubated with 1.25 μM staurosporine for 18 h [31]. Annexin V positive/7AAD-negative cells were considered to be in the early apoptotic phase, and cells positive for both Annexin V and 7AAD were considered to be late apoptotic. 

### 2.5. Detection of Intracellular ROS

Cells were seeded in 6-well plates at 3 × 10^5^ cells/well. Twenty-four hours later, cells were pre-incubated with the phytonutrients or estradiol for 24 h. The medium was then replaced with one containing the treatment compounds with or without H_2_O_2_ and incubated for an additional 90 min. Intracellular ROS were detected by 2′,7′-dichlorofluorescin diacetate (DCFH-DA, Sigma-Aldrich) staining. Briefly, cells were trypsinized, and washed by Hanks’ solution containing 10 mM HEPES buffer, pH = 7.4, followed by cell staining with 5 μM DCFH-DA for 30 min at 37 °C in the dark. Cells were analyzed by flow cytometry using the Gallios instrument. Cells treated with 0.5 mM H_2_O_2_ for 15 min were used as a positive control. Untreated and unstained cells were used as a negative control. 

### 2.6. Transient Transfection and ARE/Nrf2 Reporter Gene Assay

NHDF cells were seeded in 24-well plates at 10^5^ cells/well. Twenty-four hours later, they were transfected using jetPEI reagent (Polyplus Transfection, Illkrich, France). Briefly, cells were rinsed once with serum-free medium, followed by the addition of 0.45 mL of medium and 50 μL of a mixture containing DNA and jetPEI reagent at a charge ratio of 1:5. The total amount of DNA was 0.25 μg, containing 0.2 μg 4xARE reporter construct [32] that was kindly provided by Dr. M. Hannink (University of Missouri-Columbia, Columbia, MO, USA) and 0.05 μg Renilla luciferase (P-RL-null) expression vectors, which served as an internal transfection standard, and was purchased from Promega (Madison, WI, USA). These conditions were found to be optimal for the dermal fibroblasts. The cells were then incubated for 6 h at 37 °C. Next, the medium was replaced with DMEM-DCC-FBS plus the test compounds, and cells were incubated for an additional 16 h. Cell extracts were prepared for luciferase reporter assay (Dual Luciferase Reporter Assay System, Promega) according to the manufacturer’s instructions, and luminescence was determined using a Turner Biosystems luminometer (Sunnyvale, CA, USA).

### 2.7. Statistical Analysis

All experiments were performed in triplicate or duplicate and repeated two to seven times as indicated in the figure legends. Statistically significant differences between two experimental groups were determined using one-way ANOVA with Dunnett’s multiple comparison post-hoc analysis. Data are presented as the mean ± SEM. *p* < 0.05 was considered statistically significant. The statistical analyses were performed using GraphPad Prism 6.0 software (Graph-Pad Software, San Diego, CA, USA).

## 3. Results

### 3.1. H_2_O_2_ Induces Cell Death of Dermal Fibroblasts, Increases MMP-1 Secretion, and Decreases Pro-Collagen 1a1

Oxidative stress in human primary dermal fibroblasts was induced by incubating the cells with increasing concentrations of H_2_O_2_ (0–75 µM) for 24 h. The cell number, checked by the XTT method, decreased by H_2_O_2_ dose-dependently. At 50 µM, only about 20% of the cells were viable, and at higher concentrations, almost all cells died (Figure 1a). MMP-1 and pro-collagen 1a1 secretion were measured in the media using specific ELISA assays. H_2_O_2_ increased MMP-1 secretion and decreased pro-collagen 1a1 secretion (Figure 1b). These effects of H_2_O_2_ in dermal fibroblasts, cell death, and the inverse changes in MMP-1 and pro-collagen secretion may cause thinning of the dermis and a reduction in skin elasticity.

### 3.2. The Phytonutrients of Tomato and Rosemary Extracts and the Hormone Estradiol Protect Fibroblasts from H_2_O_2_-Induced Cell Damage

To investigate whether phytonutrients can protect NHDF cells from H_2_O_2_-induced damage, cells were pre-incubated for 24 h with increasing concentrations of tomato or rosemary extracts before exposing them to H_2_O_2_ for an additional 24 h in the presence of the phytonutrients. Partial protection from cell death was evident at most tested concentrations of the extracts (Figure 2a,b); however, these effects were not significant. Nonetheless, large and significant effects were obtained in the secretion of MMP-1 and pro-collagen, and both tomato and rosemary extracts reversed the effects of H_2_O_2_. Tomato extract reduced the H_2_O_2_-induced rise in MMP-1 secretion by about 50% (Figure 2c) and completely restored pro-collagen secretion (Figure 2e). Similarly, rosemary extract decreased MMP-1 secretion (Figure 2d) and increased pro-collagen secretion (Figure 2f) to their basal levels.

Although the protection from oxidative stress-induced damage by the tomato and rosemary extracts was significant, it was examined whether more protection can be achieved by their combination. Indeed, the combination of rosemary extract with tomato extract resulted in better protection of the cells compared to each compound alone (Figure 3). The effect of the combination on the cell number was not significantly different from the single compounds (Figure 3a); however, in these experiments, a significant reduction in MMP-1 secretion was evident only when the cells were treated with the phytonutrient combination (Figure 3b). Although the effect of single phytonutrients on pro-collagen secretion was significant, treatment with their combination induced a significantly larger increase compared to each compound alone (Figure 3c). The treatment of cells under oxidative stress, with the combination of tomato and rosemary extracts, resulted in a pro-collagen level that was higher than the basal level without H_2_O_2_.

The protective effect of estradiol on cell survival was stronger than the effect of the phytonutrients, and in the presence of 10 nM estradiol, the cells were completely resistant to cell death induced by 25 µM H_2_O_2_ (Figure 4a). Similar to the effects of the phytonutrients, estradiol decreased MMP-1 secretion (Figure 4b) and increased pro-collagen secretion back to basal levels without H_2_O_2_ (Figure 4c).

### 3.3. Tomato Extract, Rosemary Extract, and Estradiol Protect Dermal Fibroblasts from H_2_O_2_-Induced Apoptosis

It was demonstrated above that phytonutrients and estradiol protect dermal fibroblasts from H_2_O_2_-induced cell death. Consequently, it was examined whether H_2_O_2_ induced apoptosis and whether the tested compounds protected against this route of cell death. Using flow cytometry and Annexin-V/7AAD staining, it was found that H_2_O_2_ causes apoptotic cell death (Figure 5). The typical flow cytometric data of one experiment with estradiol is shown in Figure 5a. Only about 1% of cells were apoptotic when the cells were treated with the vehicle, with or without estradiol (almost all the cells appear in the unstained lower left square). After 12 h with H_2_O_2_, 10% of the cells were in an early apoptotic state (lower right panel), and 81% were in late apoptosis (upper right square). Pre-incubation with 10 nM estradiol resulted in a reduction to 8% and 44% of early- and late-apoptotic cells, respectively. The average total apoptosis (early + late) which was negligible in the control cells, increased after H_2_O_2_ treatment to more than 90% (Figure 5b). Estradiol and the phytonutrients reduced total apoptosis to less than 30%.

### 3.4. Pre-Treatment of Dermal Fibroblasts with Tomato Extract, Rosemary Extract, and Estradiol Reduces ROS Generated by H_2_O_2_

To verify that fibroblast cell damage was associated with oxidative stress, the cytoplasmic ROS level was determined by flow cytometry using a DCFH probe. Cells that were treated with H_2_O_2_ for 90 min showed a large increase in DCF fluorescence (Figure 6a, red tracing). The average geometric means of fluorescence intensities (MFI), indicating average ROS levels, increased by about six-fold (Figure 6c). Pre-incubation with estradiol (Figure 6a, black tracing) resulted in a tracing similar to the control (green tracing). ROS levels, as indicated by the MFI values, were reduced by estradiol, and the tomato and rosemary extracts back to basal level (Figure 6c). In a preliminary experiment, it was found that a reduction in ROS also occurred when the protective agents were added only during the pre-incubation, but not together with H_2_O_2_ (data not shown). This suggests that pre-incubation is required in order to reduce the ROS level. Indeed, when estradiol or the phytonutrients were added to cells at the same time as H_2_O_2_ without pre-incubation, there was no reduction in ROS levels (Figure 6d and the black tracing in Figure 6b). These results indicate that pre-incubation of the fibroblasts with the phytonutrients and estradiol is necessary to reduce these levels, and that the reduction in the ROS level did not occur by a chemical scavenging reaction between the protective agents and ROS. The increase in ROS levels by H_2_O_2_ and their reduction by the phytonutrients and estradiol was associated with cell damage and its attenuation, respectively. Thus, it was tested whether, without pre-incubation, cells can be rescued from ROS-induced death. However, when the phytonutrients or estradiol were added at the same time as H_2_O_2_, the cell number remained low and was the same as that without the protecting compounds (Figure 6e).

### 3.5. Oxidative Stress Increases the Induction of the ARE/Nrf2 Transcription System by the Phytonutrients and Estradiol

The results in the previous section suggest that pre-incubation with the phytonutrients and estradiol is required to decrease ROS levels and protect fibroblasts from H_2_O_2_-induced adverse effects. This implies that during pre-incubation, the cells increased their antioxidant defense capacity. One probable explanation for this increased capacity is the induction of the ARE/Nrf2 transcription system, which induces the transcription of antioxidant enzymes and is known to be activated by the tested compounds. The ARE/Nrf2 transcriptional activity was determined by a reporter gene assay. As expected, this transcription system was activated by rosemary extract (Figure 7a), tomato extract (Figure 7b), and estradiol (Figure 7c) from two- to four-fold in the absence of H_2_O_2_. H_2_O_2_ alone slightly increased this activity by two- to three-fold. In the presence of H_2_O_2_, the activation by the phytonutrients and estradiol was significantly higher than that of the compounds or H_2_O_2_ alone.

## 4. Discussion

The current study suggests that dietary and hormonal supplementation can prevent oxidative stress-induced damage to dermal fibroblasts. The inhibition of dermal fibroblasts cell death and the reversal of oxidative stress-induced increase in MMP-1 secretion and reduction in pro-collagen levels, that were demonstrated here, suggest that ingestion or topical application of these and similar compounds can prevent oxidative stress-induced reduction in skin depth and elasticity. The protection of dermal fibroblasts by carotenoids, polyphenols, and estradiol shown in the current study may be relevant to skin damage that is caused by several factors such as UV radiation and environmental pollutants, inducing the generation of ROS [2] (Figure 1). In this study, the dietary carotenoids, polyphenols, and estradiol reduced oxidative stress-induced apoptosis of dermal fibroblasts, reduced MMP-1 secretion, and increased pro-collagen levels. Previous studies have shown the induction of apoptosis by oxidative stress in keratinocytes [33] and fibroblast [10] skin cells. Carotenoids, including lutein [34] and astaxanthin [35], were shown to increase the keratinocyte cell number by reducing UV-induced apoptotic cell death. An in vivo study showed that the topical application of lycopene was capable of suppressing a UVB-induced cascade of apoptosis, as indicated by the reduced expression of caspase-3 in murine skin [36]. Moreover, the protective role of polyphenols in skin cells has been shown in several studies. The survival of human dermal fibroblasts under H_2_O_2_-induced oxidative stress was increased by strawberry extract [37]. Various dietary polyphenols, including rosemary extract, were shown to increase keratinocyte cell survival after UVB irradiation [38,39,40,41]. Apoptosis inhibition was evident in most of these studies [38,39,41], and one compound, delphinidin, found in pomegranates, berries, and other pigmented fruits and vegetables, was also shown to inhibit apoptosis in mice skin [38]. 

In the skin extracellular matrix, similar to other tissues, there is a balance between the synthesis of new pro-collagen and the degradation of matrix collagen. This balance is disrupted by oxidative stress, which causes an increase in the production of MMPs [11,12,42] and a reduction in pro-collagen expression [11,13] (Figure 1). Thus, the larger amount of pro-collagen attained by phytonutrient and estradiol treatments, and the reduced level of MMP-1, can restore this balance and, thus, improve skin elasticity. Previous studies have shown that polyphenols reduce MMP expression, both in vivo [43] and in vitro [17]. A similar reduction in MMP expression was shown by carotenoids in dermal fibroblasts [44] and in human skin [45,46]. Topical estradiol treatment reduced MMP-1 levels and increased pro-collagen expression in human skin [47], with similar effects found in murine bones [48]. Several signaling pathways are involved in the mechanism of oxidative stress-induced changes in MMP and collagen. A number of studies have shown that ROS induces the activation of MAPKs, which activate the NFκB and AP-1 transcription systems, leading to the increased expression of pro-inflammatory genes and MMPs [49,50,51]. Thus, inhibition of MAPKs, NFκB, and AP-1 by phytonutrients [49] may be the mechanism for reducing MMP-1 by the phytonutrients and estradiol in the current study (Figure 1). A study from our laboratory showed that carotenoid derivatives inhibit NFκB and suggested a molecular mechanism for this inhibition [52]. β-carotene was found to inhibit NFκB in cancer cells [53], and lutein inhibited this transcription system in various cells of the choroid complex of the eye, both in vivo and in vitro [54]. The polyphenol, resveratrol, inhibited NFκB activity in mouse skin by blocking IκB kinase activity [55], and a study from our laboratory demonstrated inhibition of UVB-induced NFκB activity by several carotenoids and polyphenols in human keratinocytes [21]. Estradiol was also shown to inhibit NFκB activity by increasing the expression of IƙBα, the inhibitor of NFκB [56].

The protective effects of the tested compounds in the current study were associated with a marked reduction in ROS levels only when the cells were pre-incubated with the compounds before adding H_2_O_2_. Thus, as discussed in Section 3.5, antioxidant defense was indeed improved during the pre-incubation, and this was possibly achieved by increased transcriptional activity of ARE/Nrf2, which can increase the expression of antioxidant enzymes, which can lead to ROS reduction (Figure 1). Regulation of transcription and gene expression, including ARE/Nrf2, has been found to play a significant role in the effect of phytonutrients in many cellular processes [57]. As was found in the current study, carotenoids were shown to activate ARE/Nrf2 [26,28]. This transcription system can also be activated by polyphenols such as carnosic acid [58]. ARE/Nrf2 transcriptional activity is also activated by increased ROS levels, in order to lessen oxidative stress [19,59] (Figure 1). Notably, in our study, the application of the phytonutrients and estradiol during oxidative stress increased ARE/Nrf2 activity to a level that was higher than that of the compounds or H_2_O_2_ alone. Similar to the activation of ARE/Nrf2 and the reduction in ROS in dermal fibroblasts described in the current study, estradiol activated ARE/Nrf2 and decreased ROS levels in neuronal cells, leading to cell survival and attenuation of homocysteine cytotoxicity [25]. In addition, estradiol can increase the expression of antioxidant enzymes by binding of estrogen receptor β to an estrogen response element within the promoter region of antioxidant enzymes [60] (Figure 1). An additional mechanism for the antioxidant effect of estradiol was suggested by Borrás et al. [61], who demonstrated that estradiol decreases hydrogen peroxide in isolated hepatic mitochondria, probably via interaction with specific receptors present in the mitochondria. Several publications suggest that the antioxidant effects of estradiol and other estrogens is mediated by direct chemical scavenging of ROS and not by the estrogen receptors [62]. This potential mechanism is controversial because scavenging of ROS occurs only at high, non-physiological estrogen concentrations. For example, 1 µM estradiol was required to inhibit superoxide radical production by bovine heart endothelial cells, and about 50 µM was used to inhibit lipid and low-density lipoprotein peroxidation in non-cellular experimental systems [63]. Lower estradiol concentrations (100 nM) were used to reduce ROS formation in Friedreich’s ataxia skin fibroblasts [62], a concentration that is ten times higher than the 10 nM estradiol used in the current study. Moreover, in the experimental protocol used in the current study, direct interaction between estradiol and ROS was possible only in the oxidative stress phase of the experiments, when H_2_O_2_ and estradiol were both present. However, as shown in Figure 6, the protective effect of estradiol (and the phytonutrients) depends on the pre-incubation phase, when H_2_O_2_ is not present. These results suggest that the interaction between H_2_O_2_ and estradiol probably did not occur in the current experimental protocol. Based on these considerations, it is concluded that activation of defense mechanisms during the pre-incubation phase probably explains the protective effects of estradiol and the phytonutrients in oxidative stress-challenged dermal fibroblasts. Activation of ARE/Nrf2, which was demonstrated in the current study, and inhibition of several signaling pathways such as MAPKs, NFκB, and AP-1, are possibly involved in the protection of dermal fibroblasts. 

## 5. Conclusions

This study indicates that carotenoids, polyphenols, and estradiol protect dermal fibroblasts from ROS-induced damage and suggests that a balanced diet rich in phytonutrients may improve skin health and appearance. In addition, improvement in skin health can be achieved by topical application of estradiol and crude or purified dietary compounds. Oxidative stress can result from numerous environmental factors, and also occurs during ageing. Because this stress may negatively affect many human tissues, the implications of this study may also be relevant to various other pathologies such as inflammation, neurodegenerative and cardiovascular diseases, and cancer.

## Data Availability

Data is contained within the article.

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
