# Peer review of "The Protective Effect of Carotenoids, Polyphenols, and Estradiol on Dermal Fibroblasts under Oxidative Stress"

_antioxidants, 2021, doi:10.3390/antiox10122023_

Round 1

Reviewer 1 Report

Dear Authors:

In the manuscript by Darawsha et al has demonstrated the protective effect of carotenoids, polyphenols, and estradiol on dermal fibroblasts under oxidative stress. I have just a few suggestions.

  1. Background and citations about ROS are not enough or missing in introduction. ROS plays an important role in cancer development, neurological disorders and so on. Please add more information about it. (Please cite: 1. Chen et al. Semin Cancer Biol. 2020 Oct 6:S1044-579X(20)30203-0. doi: 10.1016/j.semcancer.2020.09.012.  2. Shekhar et al. International Journal of Molecular Sciences. 2021; 22(4):2074. https://doi.org/10.3390/ijms22042074)
  2. Could authors please add a scheme figure about the potential molecular pathway of protective effect of carotenoids, polyphenols, and estradiol on dermal fibroblasts under oxidative stress, if possible?

Author Response

In the manuscript by Darawsha et al has demonstrated the protective effect of carotenoids, polyphenols, and estradiol on dermal fibroblasts under oxidative stress. I have just a few suggestions.

  1. Background and citations about ROS are not enough or missing in introduction. ROS plays an important role in cancer development, neurological disorders and so on. Please add more information about it. (Please cite: 1. Chen et al. Semin Cancer Biol. 2020 Oct 6:S1044-579X(20)30203-0. doi: 10.1016/j.semcancer.2020.09.012.  2. Shekhar et al. International Journal of Molecular Sciences. 2021; 22(4):2074. https://doi.org/10.3390/ijms22042074)

More information was added about the role of ROS in degenerative diseases with additional references (lines 36-38).

  1. Could authors please add a scheme figure about the potential molecular pathway of protective effect of carotenoids, polyphenols, and estradiol on dermal fibroblasts under oxidative stress, if possible?

A scheme was added (Scheme 1, page 12), and it is also used as a graphical abstract.

Reviewer 2 Report

A very interesting and convincing manuscript indicating that carotenoids, polyphenols, and estradiol protect dermal fibroblasts from ROS-induced damage and suggests that a balanced diet rich in phytonutrients may improve skin health and appearance. Oxidative stress occurs during ageing but also may negatively affect many human tissues, so the implications of this study may also be relevant to various other pathologies such as inflammation, neurodegenerative and cardiovascular diseases, and cancer. It has been shown that estrogens, carotenoids and polyphenols activate the cell’s antioxidant defense system by increasing antioxidant response element/Nrf2 (ARE/Nrf2) transcriptional activity and reducing the inflammatory response. Research has also been conducted on cell death, apoptosis secretion of MMP-1, and pro-collagen secretion. It is true that the used compounds (including the main components of the extracts) are known and have documented biological activity, but it is useful to study their influence together as it allowed to obtain answers to their mutual influence on the final biological effect.

Information on the used tomato extract (LycomatoTM) and rosemary extract (a gift of Lycored Ltd., Beer Sheva, Israel) requires additional supplementation. It is necessary to provide the full composition of the extracts or to indicate other, commercially available extracts with such a composition that it is possible to obtain full repeatability of the obtained results.

Author Response

Reviewer 2

More information was added about the tomato extract, which is a branded commercial product (LycomatoTM). Most of these additional components are monoacylglycerols, triacylglycerols, and phospholipids, which are assumed to be inactive in the concentrations used in this study. The composition of the rosemary extract was only partially determined, and we specified this in the text. The corrected description of both products is found in lines 79-86.

Reviewer 3 Report

This paper aims at investigating the protective effect dietary-derived compounds (tomato and rosemary extracts) and estradiol on normal human dermal fibroblasts under oxidative stress from hydrogen peroxide. The cells were examined in terms of cell number and apoptosis, while MMP-1 and pro-collagen secretion were used as markers of "skin damage". The cells were pre-treated either with the protective agents separately or in combination of tomato and rosemary extracts. Further, the pre-treatment protocol was compared with a protocol excluding pre-incubation with the protective agents. Finally, the effect of pre-treatment with the phytonutrients or estradiol on the ARE/Nrf2 transcriptional activity was determined by a reporter gene assay.

Overall, the experimental design and rationale behind the chosen techniques and methods are clearly explained and justified in a scientifically sound manner. However, some minor comments follow, which in my opinion should be clarified before publication.

The fact that the experiments do not cover the effects of pre-treatment with the phytonutrients or estradiol, followed by oxidative stress (i.e., induced by H2O2) without the protective agents present should be motivated. For example, is it possible that H2O2 interacts chemically with the protective agents, leading to reduced oxidative stress? It is recommended to highlight this fact throughout the paper. In particular, the ARE/Nrf2 transcription system was shown to be activated by rosemary and tomato extracts and estradiol, as well as H2O2 alone. Based on this, the authors suggest that the protective mechanism is due to increased expression of antioxidant enzymes, which leads to reduced oxidative stress. However, this is not supported by any experiments showing upregulation of, e.g., catalase.  

What was the reason for not combining all agents (both phytonutrients and estradiol) in the combined pre-treatment protocol?

The authors use the rather general term "skin cells" in several occasions. Please, be more specific for each case; what cell type is meant and where in the skin organ is it expected to be found.

The authors mainly discuss dietary sources of phytonutrients and estradiol. What about topical application of these protective agents? Is this a potential route of application or is the dermal fibroblast cell type better targeted via systemic delivery?

Lines 40-42: This sentence should be supported by references specifying what epidemiological and clinical studies the authors refer to. In general, the authors should make sure that given claims are supported by references (e.g., skin health is impaired in menopause due to reduction in estradiol levels, etc.).

Author Response

The author's comments were divided and numbered to allow a separate response for each comment:

  1. The fact that the experiments do not cover the effects of pre-treatment with the phytonutrients or estradiol, followed by oxidative stress (i.e., induced by H2O2) without the protective agents present should be motivated. For example, is it possible that H2O2interacts chemically with the protective agents, leading to reduced oxidative stress? It is recommended to highlight this fact throughout the paper.

As stated by the reviewer, the option of "pre-treatment with the phytonutrients or estradiol, followed by oxidative stress (i.e., induced by H2O2) without the protective agents was not presented in the manuscript. One such experiment was performed, and when the ROS level was measured, we found a complete reduction in ROS level by 10 µM rosemary extract and a more than 90% reduction by 10 nM estradiol. Since the results were similar to those done using the regular protocol (including protective agents during the oxidative stress phase, Figure 6 a, c), we did not continue with experiments without the protective agents in the oxidative stress phase. A sentence describing this preliminary experiment was added in Section 3.4 (data not shown, lines 278-281). The answer to the reviewer's question: "is it possible that H2O2 interacts chemically with the protective agents, leading to reduced oxidative stress?" may be theoretically positive, but in that case, if the protective agent's concentration is enough to reduce H2O2 chemically, then ROS had to be reduced in the experiments without pre-incubation, where H2O2 was added together with the protective agents. However, as shown in Figure 6, this did not occur, thus, suggesting that a chemical scavenging reaction between the protective agents and ROS did not occur under the condition of these experiments. A sentence with this conclusion was added in Section 3.4 (lines 285-286). Additional information in this respect, specific to estradiol, is found in the response to comment 10 of Reviewer 4.

  1. In particular, the ARE/Nrf2 transcription system was shown to be activated by rosemary and tomato extracts and estradiol, as well as H2O2alone. Based on this, the authors suggest that the protective mechanism is due to increased expression of antioxidant enzymes, which leads to reduced oxidative stress. However, this is not supported by any experiments showing upregulation of, e.g., catalase.  

Adding results about the levels of antioxidant enzymes, as suggested by the reviewer, would certainly support our conclusions. However, we (reference 26), and many others, have shown a correlation between activation of the ARE/Nrf2 transcription system and the expression of phase 2 and antioxidant enzymes. Therefore, showing this in the current manuscript, although it would strengthen the conclusions, it would not add a new perception, and thus, we did not include such experiments here.

  1. What was the reason for not combining all agents (both phytonutrients and estradiol) in the combined pre-treatment protocol?

We did plan to carry out the suggested experiments; however, the effects of estradiol were very strong and resulted in almost a complete reversal of the H2O2 effects (Figure 4). These effects were not dose dependent, so that we were not able to get lower effects of estradiol alone. Because cooperative or synergistic effects can be seen only if the effects of individual compounds are not too strong, it was not possible to show even additive effects of estradiol with the phytonutrients. 

  1. The authors use the rather general term "skin cells" in several occasions. Please, be more specific for each case; what cell type is meant and where in the skin organ is it expected to be found.

The term "skin cells" was replaced by "dermal fibroblasts" or "keratinocytes" in the appropriate places (lines 18, 62, 74 and 404). The location in the skin of keratinocytes in the epidermis (line 62) and dermal fibroblasts in the dermis (line 68-69) was added to the Introduction section.

  1. The authors mainly discuss dietary sources of phytonutrients and estradiol. What about topical application of these protective agents? Is this a potential route of application or is the dermal fibroblast cell type better targeted via systemic delivery?

Topical application of these protective agents is indeed a potential route, and in the discussion, we did cite several papers describing results using this route (references 36, 47). In addition to these citations, "topical application" was mentioned in the beginning of the discussion (line 335-336) and in the conclusions (lines 435-436)

  1. Lines 40-42: This sentence should be supported by references specifying what epidemiological and clinical studies the authors refer to. In general, the authors should make sure that given claims are supported by references (e.g., skin health is impaired in menopause due to reduction in estradiol levels, etc.).

References 14-19 were added in the paragraph mentioned by the reviewer (lines 44-52).  Regarding the sentence "skin health is impaired in menopause….." – the related reference [18] appeared just before this phrase, after the first half of the sentence (line 55). Due to the possible confusion, the reference was also added at the end of the sentence (line 56).

Reviewer 4 Report

This study looks at dermal fibroblasts and how they can be protected from oxidative stress. The study is well designed and experiments are performed with adequate methodology. Most notably, protective effects by estradiol, carotenoid, and polyphenolic compound appeared only after a 24h preincubation underlining a mechanistic response on the level of cellular ROS defense. Furthermore, H2O2 stress and pretreatment with the compounds appear to have an additive effect on ARE/Nrf2 induction. I liked reading the study and I compliment the authors on their work.

There are some minor additions and changes I suggest, nothing mayor:

line 71 Please provide some more information on how these extracts are made.

line 135 How can you be sure the two independent plasmids can serve as a transfection control for each other? Please add more explanation. Provide more details on the 4xARE reporter construct or at least a reference.

Provide more details on the treatment you applied in the groups named "vehicle" - did you combine THF and ethanol? 
If so, did you account for any additive effects?

Explain DCC-FBS - why does one use it.

Figure 1b - consider using two more distinct colors - red and orange are hard to discern and impossible to discern for color-impaired colleagues...

Figure 2: two biological replicates are poor, can you do more replicates? If this is not possible, please clearly state the group that only had 2 biological replicates in the figure text. 

Figure 3: unfavorable color choices again. Please improve readability. Also, there seems to be a mistake: rosemary extract is supposed to be gray - in the figure it is orange...

Figure 6d: please improve the figure text - does this mean the 3 compounds were added with the H2O2 treatment? This is not entirely clear to me...

Figure 7: Again I am not convinced by your color scheme - since there is no black bar the connection to the colored bars is not entirely logical. Again orange and red are hard to discern for some fraction of the population. 

Several studies exist that state direct oxidation of Estradiol by ROS, or the ability of other lipids to buffer oxidative stress - I would suggest adding a sentence or two on these observations in your introduction or discussion. Have a look at e.g.

https://www.sciencedirect.com/science/article/pii/S092544390900218X
https://www.thelancet.com/journals/lancet/article/PIIS0140-6736(94)91117-7/fulltext

Can you rephrase/reposition the first sentence in the discussion? "The current study suggests that dietary and hormonal supplementation can prevent oxidative stress-induced skin damage and may restore skin depth and elasticity." Since you did not look at skin damage, skin depth or elasticity directly, in my opinion, this should be the last sentence in your discussion, and everything before should lead up to this conclusion...

What I am missing in the discussion is a paragraph about the application of these three compounds - some references you state applied phytochemicals topical - so how do you see this with the three present compounds - do they have to be ingested or could you think of topical application.

The English language is really flawless and I found no typos so far...

Author Response

Reviewer 4

This study looks at dermal fibroblasts and how they can be protected from oxidative stress. The study is well designed and experiments are performed with adequate methodology. Most notably, protective effects by estradiol, carotenoid, and polyphenolic compound appeared only after a 24h preincubation underlining a mechanistic response on the level of cellular ROS defense. Furthermore, H2O2 stress and pretreatment with the compounds appear to have an additive effect on ARE/Nrf2 induction. I liked reading the study and I compliment the authors on their work.

There are some minor additions and changes I suggest, nothing mayor: The comments were numbered to make it easier to follow our responses,

  1. line 71 Please provide some more information on how these extracts are made.

This information was added (lines 79-84) together with the information about the composition of the extract that was requested by Reviewer 1.

  1. line 135 How can you be sure the two independent plasmids can serve as a transfection control for each other? Please add more explanation. Provide more details on the 4xARE reporter construct or at least a reference.

A reference ([32], Cullinan et al. 2003) was added to the 4xARE reporter construct (line 147). The use of Renilla luciferase as a transfection control in reporter gene assays is a widely used and accepted method. We titrated the relative amounts of the two plasmids and the transfection reagent for the cells used in the current study to achieve optimal conditions. We referred to this in Section 2.6 line 151.

  1. Provide more details on the treatment you applied in the groups named "vehicle" - did you combine THF and ethanol? 
    If so, did you account for any additive effects?

Added in Section 2.3 lines105-107

  1. Explain DCC-FBS - why does one use it.

The explanation was provided in the following sentence in Section 2.2: " Before each experiment, cells were depleted of steroid hormones by maintaining them for 3–5 days in phenol red-free DMEM supplemented with 10% DCC-FBS (DMEM-DCC-FBS)". We added another sentence (lines 100-101) to make this clearer.

  1. Figure 1b - consider using two more distinct colors - red and orange are hard to discern and impossible to discern for color-impaired colleagues... These colors were changed.
  2. Figure 2: two biological replicates are poor, can you do more replicates? If this is not possible, please clearly state the group that only had 2 biological replicates in the figure text. 

Out of the 62 experimental groups in Figure 2, seven were repeated in only two experiments. To clearly indicate these groups, we added a mark (©) in the center of the relevant columns and explained it in the figure text (lines 205-206). It should be stressed that in spite of having only two repetitions, the results of six of these groups were significantly different from the relevant control.

  1. Figure 3: unfavorable color choices again. Please improve readability. Also, there seems to be a mistake: rosemary extract is supposed to be gray - in the figure it is orange... These colors were changed.
  2. Figure 6d: please improve the figure text - does this mean the 3 compounds were added with the H2O2 treatment? This is not entirely clear to me...

To clarify this issue, we now wrote that H2O2 was added together with the treatment compounds (lines 296-297). 

  1. Figure 7: Again I am not convinced by your color scheme - since there is no black bar the connection to the colored bars is not entirely logical. Again orange and red are hard to discern for some fraction of the population. The colors were changed.
  2. Several studies exist that state direct oxidation of Estradiol by ROS, or the ability of other lipids to buffer oxidative stress - I would suggest adding a sentence or two on these observations in your introduction or discussion. Have a look at e.g.
  3. https://www.sciencedirect.com/science/article/pii/S092544390900218X
  4. https://www.thelancet.com/journals/lancet/article/PIIS0140-6736(94)91117-7/fulltext

The first paper suggested by the reviewer proposes a receptor-mediated antioxidant effect of estradiol directly in the mitochondria. This possible mechanism was added to the Discussion (lines 408-411).

As to the reviewer's suggestion about " direct oxidation of Estradiol by ROS": Direct chemical scavenging of ROS by estradiol is possible. However, we should emphasize that although estradiol can interact directly with ROS (e.g., Richardson 2011), such scavenging occurs only at high, non-physiological concentrations of estradiol, which were not used in the current study. For example, in one paper (Ayres 1998), very high concentrations (30-50 µM) were required for such activity in both cellular and acellular systems. In another paper (Richardson 2011), lower estradiol concentrations were used, but to achieve significant effects, the concentration had to be 10 to 100 times higher than the 10 nM estradiol used in the current study. In addition, direct interaction between estradiol and ROS was possible in the oxidative stress phase of the experiments, when H2O2 and estradiol were both present. However, we have shown in Figure 6 that the protective effect of estradiol (and the phytonutrients) depends on the pre-incubation phase, when H2O2 is not present. These results suggest that the interaction between H2O2 and estradiol probably did not occur in this experimental protocol. This issues were added to the Discussion (lines 411-426).

  1. Can you rephrase/reposition the first sentence in the discussion? "The current study suggests that dietary and hormonal supplementation can prevent oxidative stress-induced skin damage and may restore skin depth and elasticity." Since you did not look at skin damage, skin depth or elasticity directly, in my opinion, this should be the last sentence in your discussion, !!! and everything before should lead up to this conclusion...

We changed the first sentence from prevention of "skin damage" to prevention of "damage to dermal fibroblasts" (line 331) which is what we presented in this study. The phrase about "skin depth and elasticity" was moved to the next sentence where it is linked to the studied effects in dermal fibroblasts (lines 334-336).   

  1. What I am missing in the discussion is a paragraph about the application of these three compounds - some references you state applied phytochemicals topical - so how do you see this with the three present compounds - do they have to be ingested or could you think of topical application.

As suggested by the reviewer (and by Reviewer 3), and as stated in some of the references (36, 47), topical application is a potential route of administration of the compounds. To emphasize this, "topical application" was discussed in the beginning of the Discussion (lines 335-336) and to the conclusions (line 435-436).

Round 2

Reviewer 1 Report

Strongly suggest to publish.